# The effect of coronary stent policies on the risk of percutaneous coronary intervention among acute coronary syndrome patients in Shanghai: Real-world evidence

Zhenyi Shao[1], Dongzhe Lu[2], Yue Wang[1], Di Xue[2]*

1 Health Statistics Department, Shanghai Statistics Center for Health, Shanghai, People's Republic of China,
2 School of Public Health, NHC Key Laboratory of Health Technology Assessment (Fudan University), Fudan University, Shanghai, People's Republic of China

* xuedi@shmu.edu.cn

## Abstract

### Objective

This study aimed to analyze the effect of coronary stent policies implemented in Shanghai on the risk of percutaneous coronary intervention (PCI) in acute coronary syndrome (ACS) inpatients based on real-world data.

### Methods

Two retrospective cohorts of inpatients with a first diagnosis of ACS who had undergone PCI for the first time in the previous year in Shanghai hospitals were examined (one for the postpolicy period and the other for the prepolicy period). $\chi^2$ tests were used to compare categorical variables between the two cohorts. Single- and multivariate Cox proportional hazards models were used to compare the risk of major adverse cardiovascular events (MACEs) between the two cohorts.

### Results

A total of 31,760 ACS patients were included in this study. The proportion of ACS inpatients who had at least one bid-winning stent and 3 or more coronary stents implanted for first-time PCI in the postpolicy cohort was higher than that in the prepolicy cohort (86.52% vs. 55.67% and 6.27% vs. 4.39%, respectively; all p values < 0.0001). The single- and multivariate Cox proportional hazards models revealed that the unadjusted and adjusted hazard ratios for MACEs at 1 year after PCI for the postpolicy cohort relative to the prepolicy cohort were 0.869 (P<0.0001) and 0.814 (P = 0.0007), respectively.

### Conclusions

The implementation of coronary stent policies changed coronary stent utilization but had no significant adverse effects on the risk of PCI among ACS patients in Shanghai in the short

**Data Availability Statement:** All data underlying the findings described in our manuscript were from the Shanghai Statistics Center for Health. According to Article 33 in "Shanghai Statistical

Regulations" issued in 2016, the Shanghai Statistics Center for Health cannot externally provide or disclose individual information, or information that could identify individuals or infer individual identity. In the study, we analyzed the data in the Shanghai Statistics Center for Health and the analysis results were sent to us after they were checked by the Shanghai Statistics Center for Health to ensure no individual information were included. Therefore, all data underlying the findings described in our manuscript cannot be freely available to other researchers. In addition, after the study protocol was approved by the IRB of School of Public Health, Fudan University and the Shanghai Statistics Center for Health, the data utilization for the study was monitored by the Shanghai Statistics Center for Health. The Shanghai Statistics Center for Health has no specific department or committee (such as a data access committee, ethics committee, or other institutional body) that is responsible to the management of data utilization. However, the departments that use the data will be responsible for proper data utilization. For any supervision purpose, the IRB of School of Public Health, Fudan University or the relevant government agencies of China can access our research data in the Shanghai Statistics Center for Health.

**Funding:** The author(s) received no specific funding for this work.

**Competing interests:** The authors have declared that no competing interests exist.

term. However, the reasons for changes in the number of coronary stents implanted should be analyzed and addressed in the future.

## Introduction

Coronary heart disease (CHD) is a leading cause of death worldwide, including in China [1]. Acute coronary syndrome (ACS) is the most serious type of CHD and has rapid onset and high mortality. Percutaneous coronary intervention (PCI) is a common therapy for CHD and is the preferred therapy for ACS [2, 3]. However, the high cost of implanted stents results in a high financial burden for patients who undergo PCI [4, 5].

To reduce the price of coronary stents to a reasonable level, the central government of China launched a national centralized volume-based procurement policy for coronary stents, and 10 products were ultimately selected as bid-winning coronary stents [6, 7]. As a result, the price of coronary stents decreased from the previous average market price of approximately 13,000 yuan each to the average bid-winning price of 670 yuan [8]. The National Healthcare Security Administration of China issued supportive measures afterward, indicating that bid-winning coronary stents would be paid at the bid-winning price and be covered in the scope of basic medical insurance [9]. In addition, in January 2021, the government of Shanghai implemented relevant policies, mandating that bid-winning coronary stents should be given top priority for procurement and utilization and that the payment for prices of bid-nonwinning coronary stents should not exceed 7000 yuan [10, 11]. These national centralized volume-based procurement policies and related supportive measures for coronary stents (abbreviated as "coronary stent policies" hereafter) have been implemented in Shanghai since January 20, 2021.

Since 2018, China has implemented similar national centralized volume-based procurement policies for medicines, and previous studies have reported positive impacts of these policies on medication use, medical expenses and the accessibility of medicines [12–14]. Similar to the implementation of national centralized volume-based procurement policies for medicines in China, the implementation of coronary stent policies may reduce the medical cost borne by ACS patients who undergo PCI; however, its impact on the risk of PCI in ACS patients is unknown.

### Objectives

The present study aimed to analyze the effect of coronary stent policies implemented in Shanghai on the risk of PCI in ACS inpatients based on real-world data. This study provides evidence for future policy-making for coronary stents in China as well as in other countries.

### Methods

#### Participants

We constructed two retrospective cohorts of inpatients with a first diagnosis of ACS who had undergone first-time PCI in the previous year in Shanghai hospitals: one for those discharged from January 20, 2021, to April 30, 2022 (postpolicy), and one for those discharged from March 1, 2019, to January 19, 2021 (prepolicy). The inclusion criteria of the study were as follows: 1) inpatients with a first diagnosis of ACS who had undergone first-time PCI in the previous year in hospitals in Shanghai and were discharged between March 1, 2019, and April 30,

2022; and 2) inpatients who were Shanghai residents and by Aug 13, 2022, 1 year, at least, had passed after their first-time PCI.

In this study, ACS patients who had undergone first-time PCI in the previous year in Shanghai hospitals and had no detailed records of outpatient visits for cardiovascular disease (CVD) after first-time PCI were excluded. This exclusion criterion was set to ensure that it was possible to collect relevant information on ACS patients after PCI.

In this study, the ACS included unstable angina pectoris (International Classification of Diseases, Tenth Revision (ICD-10) code I20.0), non-ST-segment elevation myocardial infarction (MI) (ICD-10 code I21.4), ST-segment elevation myocardial infarction (ICD-10 code I21.0-.3), and others that were unable to determine a specific type (ICD-10 code I24.8 or I22.8) [15].

## Real-world data collection

All real-world data used in this study were collected from the databases of the Shanghai Statistics Center for Health and included patient characteristics, inpatient care in which first-time PCI was performed, outpatient care and readmission care during the 1 year after PCI, and health outcomes at 1 year after PCI.

The main data related to patient characteristics were age, sex, and basic medical insurance status. The main data related to inpatient care at the time of the initial PCI were medical history, admission hospital level, department, primary diagnosis, New York Heart Association (NYHA) or Killip functional classification, number and type of coronary stent implanted in PCI, and admission and discharge dates. The main data related to outpatient care during the 1 year after PCI were visit department, diagnosis, medication prescription, and visit date; The main data related to readmission care during 1 year after PCI were admission department, primary diagnosis, PCI, coronary artery bypass grafting (CABG), and admission and discharge dates.

The medications analyzed in this study included anticoagulant and antiplatelet drugs (aspirin, clopidogrel, and ticagrelor), nitrates (nicorandil and isosorbide dinitrate), beta-blockers (metoprolol), angiotensin receptor-enkephalinase inhibitors (sacubitril/valsartan), anginal attack preventive drugs (trimetazidine dihydrochloride), cholesterol absorption inhibitors (ezetimibe), and statins (atorvastatin, rosuvastatin, and pravastatin sodium) (12 medicines in total). These 12 medicines were the most frequently prescribed at outpatient visits among ACS patients after PCI, accounting for 73.49% of the total prescriptions. Because vascular endothelial injury during PCI leads to stent thrombosis, dual antiplatelet therapy (DAPT) with aspirin plus a $P2Y_{12}$ inhibitor is the standard-of-care treatment for patients undergoing PCI [16]. According to the Chinese guidelines for cardiovascular disease prevention (2017), aspirin plus clopidogrel were recommended for use for at least 12 months after PCI; for patients who cannot tolerate clopidogrel or have shown clear evidence of clopidogrel resistance, ticagrelor or prasugrel were recommended as alternative medicines [17]. However, there were no ACS patients who used prasugrel after PCI in our study; therefore, it was not included in the analyses.

## Outcomes

We used the occurrence of major adverse cardiovascular events (MACEs) at 1 year after PCI to assess the risk of PCI in ACS patients. In this study, MACEs were defined as the composite of coronary artery restenosis or stent thrombosis ("restenosis or stent thrombosis", for short), nonfatal myocardial infarction with no revascularization ("MI", for short), coronary artery revascularization (i.e., "revascularization"), and death from any cause ("death").

## Statistical analyses

This study employed $\chi^2$ tests to compare patient characteristics, inpatient care during which first-time PCI was performed, medications prescribed during outpatient care within 1 year after PCI, and MACE rates at 1 year after PCI between the post- and prepolicy cohorts. Single- and multivariate Cox proportional hazards models ("Cox models" for short) were used to analyze the effect of the policies on the risk of MACEs after PCI. In the models, the occurrence of MACEs within 1 year was the endpoint event, and the day after PCI, when MACEs occurred, was the time point. Hazard ratios were used to estimate the relative risks between the post- and prepolicy cohorts. According to single- and multivariate Cox models, the unadjusted and adjusted hazard ratios were calculated, and MACE-free survival curves were plotted.

According to the multivariate Cox models, patient characteristics (age, sex, having or not having basic medical insurance), clinical characteristics in inpatient care during which first-time PCI was performed (medical history, admission hospital level, NYHA or Killip functional classification, and number of coronary stents implanted in the PCI), and medication used in outpatient care within 1 year after PCI were used as independent variables.

SAS 9.4, a software package from SAS Institute, Inc., Cary, NC, USA, was used for statistical analysis in this study.

This study received institutional review board approval (IRB) from the School of Public Health, Fudan University (IRB#2022-01-0937 and IRB#2022-01-0937-s). Informed consent from the patients was exempted by the IRB because this was a retrospective study with no more than minimal risk to patients. Furthermore, this study is impractical without the waiver of informed consent, and the research has important social value. The data were accessed for research purposes from July 15, 2022, to June 30, 2023, and anonymous data were used during the study analyses.

## Results

### Characteristics of ACS patients in the post- and prepolicy cohorts

A total of 31,760 ACS inpatients were retrospectively included in this study. Among them, 28.28% were in the postpolicy cohort, and 71.72% were in the prepolicy cohort. There were significant differences in age, medical insurance status, medical history (diabetes, hypertension, hyperlipidemia, stroke, chronic kidney disease, chronic obstructive pulmonary disease [COPD], MI, or previous CABG), clinical characteristics (admission hospital level and NYHA or Killip functional classification), inpatient care for first-time PCI, and most of the medicines used 1 year after PCI between the post- and prepolicy cohorts. However, there were no differences in diagnostic classification of ACS (unstable angina pectoris, ST-elevation MI, non ST-elevation MI or undetermined), patient sex or aspirin utilization between the two cohorts (Table 1).

### Coronary stent utilization in the postpolicy and prepolicy cohorts

The present study showed that the proportion of ACS patients who had 3 or more coronary stents implanted during their first PCI in the postpolicy cohort was 6.28%, which was higher than that in the prepolicy cohort (4.39%), and the proportion of ACS inpatients who had at least one bid-winning stent implanted in the postpolicy cohort was 86.52%, which was much higher than that in the prepolicy cohort (55.67%) (Table 2). In the ACS patients who underwent first-time PCI, all the bid-winning stents implanted were drug-eluting or drug-coating stents, while the bid-nonwinning stents included drug-eluting, drug-coating, and biodegradable stents. Bare metal stents were not used in any of the patients. The proportions of drug-

**Table 1. Characteristics and medications of ACS inpatients who had undergone PCI.**

| Characteristics and medications | Postpolicy (n = 8982) | Prepolicy (n = 22778) | $\chi^2$ | p value |
|---|---|---|---|---|
| **Diagnostic classification** | | | | |
| Unstable angina pectoris | 3933(43.79%) | 9691(42.55%) | | |
| Non-ST-segment elevation MI | 2054(22.87%) | 5351(23.49%) | 4.61 | 0.2026 |
| ST-segment elevation MI | 2966(33.02%) | 7649(33.58%) | | |
| Undetermined | 29(0.32%) | 87(0.38%) | | |
| **Age, years** | | | | |
| <50 | 925(10.30%) | 2097(9.21%) | | |
| 50- | 1587(17.67%) | 4082(17.92%) | | |
| 60- | 3209(35.73%) | 8323(36.54%) | 28.56 | < .0001 |
| 70- | 2343(26.09%) | 5600(24.59%) | | |
| 80- | 918(10.21%) | 2676(11.74%) | | |
| **Male** | 6786(75.55%) | 17223(75.61%) | 0.01 | 0.9038 |
| **Insurance coverage** | 7935(88.34%) | 20310(89.16%) | 4.42 | 0.0356 |
| **Hospital level** | | | | |
| Tertiary hospital | 7984(88.89%) | 20788(91.26%) | 42.62 | < .0001 |
| Secondary hospital | 998(11.11%) | 1990(8.74%) | | |
| **Previous medical history** | | | | |
| Diabetes | 2783(30.98%) | 7511(32.97%) | 11.65 | 0.0006 |
| Hypertension | 5271(58.68%) | 14451(63.44%) | 61.98 | < .0001 |
| Hyperlipidemia | 2742(30.53%) | 7638(33.53%) | 26.43 | < .0001 |
| Stroke | 81(0.90%) | 265(1.16%) | 4.09 | 0.0431 |
| Chronic kidney disease | 844(9.40%) | 2391(10.50%) | 8.53 | 0.0035 |
| COPD | 251(2.79%) | 764(3.35%) | 6.52 | 0.0107 |
| MI | 646(7.19%) | 2028(8.90%) | 24.46 | < .0001 |
| CABG | 264(2.94%) | 1002(4.40%) | 35.87 | < .0001 |
| **NYHA or Killip functional classification** | | | | |
| I | 2492(60.59%) | 4350(55.68%) | | |
| II or III | 1499(36.45%) | 3283(42.03%) | 37.08 | < .0001 |
| IV | 122(2.96%) | 179(2.29%) | | |
| **Medicines prescribed at outpatient visits during 1 year after PCI** | | | | |
| Aspirin | 6255(69.64%) | 15739(69.10%) | 0.89 | 0.3459 |
| Clopidogrel | 4691(52.23%) | 12246(53.76%) | 6.10 | 0.0135 |
| Metoprolol | 4777(53.18%) | 13051(57.30%) | 44.24 | < .0001 |
| Ticagrelor | 4037(44.95%) | 11365(49.89%) | 63.17 | < .0001 |
| Atorvastatin | 5392(60.03%) | 14113(61.96%) | 10.10 | 0.0015 |
| Ezetimibe | 1556(17.32%) | 2799(12.29%) | 138.04 | < .0001 |
| Sacubitril/valsartan | 2342(26.07%) | 2210(9.70%) | 1406.29 | < .0001 |
| Trimetazidine dihydrochloride | 443(4.93%) | 1918(8.42%) | 113.91 | < .0001 |
| Pravastatin sodium | 750(8.35%) | 2387(10.48%) | 32.81 | < .0001 |
| Nicorandil | 502(5.59%) | 694(3.05%) | 114.86 | < .0001 |
| Isosorbide dinitrate | 1409(15.69%) | 4260(18.70%) | 39.94 | < .0001 |
| Rosuvastatin | 2296(25.56%) | 4495(19.73%) | 130.17 | < .0001 |
| Others | 541(6.02%) | 1175 (5.16%) | 9.42 | 0.0021 |

**Table 2. Comparison of coronary stent utilization in the post- and prepolicy cohorts.**

| Stents implanted | | Postpolicy | Prepolicy | $\chi^2$ | p value |
|---|---|---|---|---|---|
| **At least one bid-winning stent** | | | | | |
| | Yes | 6086(86.52%) | 8950(55.67%) | 2049.02 | < .0001 |
| | No | 948(13.48%) | 7127(44.33%) | | |
| **Number of coronary stents implanted** | | | | | |
| | 1 | 4244(73.96%) | 7897(74.36%) | | |
| | 2 | 1134(19.76%) | 2257(21.25%) | 35.57 | < .0001 |
| | 3 | 290(5.06%) | 402(3.79%) | | |
| | 4 or more | 70(1.22%) | 66(0.60%) | | |

eluting stents, drug-coated stents, biodegradable stents and unknown types of stents implanted were 78.90%, 19.75%, 0.29% and 1.06%, respectively, in the first-time PCI in the postpolicy cohort, while these proportions were 71.07%, 21.90%, 4.51% and 2.52%, respectively, in the prepolicy cohort. More drug-coated stents and biodegradable stents were used the prepolicy cohort ($\chi^2$ = 600.69, p<0.0001).

## Hazard ratios of MACEs 1 year after PCI in the post- and prepolicy cohorts

The single-variate Cox models showed that the hazard ratios of total MACEs and revascularization for the postpolicy cohort relative to the prepolicy cohort were 0.869 (p<0.0001) and 0.873 (p<0.0001), respectively, while the hazard ratios of restenosis or stent thrombosis, MI and death were 1.146 (p>0.05), 0.858 (p>0.05) and 0.743 (p>0.05), respectively (Table 3).

After multivariate Cox models were used to control for other influencing factors (patient characteristics, clinical characteristics in inpatient care for first-time PCI, and medications used in outpatient care 1 year after PCI), the adjusted hazard ratios of total MACEs and revascularization for the postpolicy cohort relative to the prepolicy cohort were 0.814 (p = 0.0007) and 0.822 (p = 0.0024) (Tables 3 and 4), respectively, while the adjusted hazard ratios of restenosis or stent thrombosis, MI and death were 1.329 (p>0.05), 0.753 (p>0.05) and 0.710 (p>0.05), respectively (Table 4 and S1–S3 Tables).

Cox models were used to construct MACE-free survival curves, which demonstrated that the majority of MACEs that occurred within 1 year after PCI in ACS patients occurred within 60 days after PCI (Fig 1).

## Influences of other factors on hazard ratios of MACEs 1 year after PCI

Adopting multivariate Cox models, this study showed that male ACS patients or ACS patients with diabetes or hypertension and ACS patients who had undergone PCI at tertiary hospitals or used ticagrelor, sacubitril/valsartan and/or isosorbide dinitrate in outpatient care 1 year after PCI had a significantly higher risk of MACEs 1 year after PCI. However, ACS patients who had previously undergone CABG, had moderate cardiac impairment (NYHA or Killip functional classification II or III) in inpatient care for first-time PCI, or used aspirin and/or metoprolol in outpatient care 1 year after PCI had a significantly lower risk of MACEs 1 year after PCI (Table 4).

In addition, after controlling for other factors, ACS patients with hypertension or severe cardiac impairment (NYHA or Killip functional classification IV) had a higher risk of death 1 year after PCI, while ACS patients who had hyperlipidemia or used aspirin and/or metoprolol had a lower risk of death 1 year after PCI (S1 Table).

**Table 3. The effect of the policies on the risk of PCI at 1 year using Cox models.**

| MACEs | | Estimate | SD | $\chi^2$ | p value | Hazard ratio (95% CI) |
|---|---|---|---|---|---|---|
| **Single-variate Cox models [a]** | | | | | | |
| | Restenosis or stent thrombosis | 0.1360 | 0.3220 | 0.18 | 0.6727 | 1.146(0.610, 2.154) |
| | MI | -0.1528 | 0.0948 | 2.60 | 0.1070 | 0.858(0.713, 1.034) |
| | Revascularization | -0.1355 | 0.0318 | 18.20 | < .0001 | 0.873(0.821, 0.929) |
| | Death | -0.2971 | 0.2758 | 1.16 | 0.2813 | 0.743(0.433, 1.276) |
| | Total MACEs | -0.1404 | 0.0298 | 22.16 | < .0001 | 0.869(0.820, 0.921) |
| **Multivariate Cox models [b]** | | | | | | |
| | Restenosis or stent thrombosis | 0.2841 | 0.7185 | 0.16 | 0.6926 | 1.329(0.325, 5.433) |
| | MI | -0.2837 | 0.2020 | 1.97 | 0.1603 | 0.753(0.507, 1.119) |
| | Revascularization | -0.1964 | 0.0646 | 9.24 | 0.0024 | 0.822(0.724, 0.933) |
| | Death | -0.3426 | 0.4734 | 0.52 | 0.4693 | 0.710(0.281, 1.795) |
| | Total MACEs | -0.2060 | 0.0609 | 11.45 | 0.0007 | 0.814(0.722, 0.917) |

[a] The implementation of coronary stent policies was used in the Cox models as an independent variable (1: yes, 0: no).

[b] The implementation of coronary stent policies, patient demographic characteristics (age, sex, having medical insurance), clinical characteristics (medical history, admission hospital level, NYHA or Killip functional classification, and number of coronary stents implanted), and medicines used in outpatient care 1 year after PCI were used in the Cox models as independent variables; all independent variables were 1–0 variables (1: yes, 0: no).

## Discussion

### Changes in coronary stent utilization after policy implementation

The coronary stent policies implemented in Shanghai have considerably reduced both the bid-winning and bid-nonwinning prices of coronary stents, and bid-winning coronary stents have been given top priority for procurement and utilization and are covered by basic medical insurance [8, 10, 11]. These policies have led to changes in coronary stent utilization among ACS patients who undergo PCI. Our study revealed that the proportion of ACS patients who had at least one bid-winning stent implanted in the postpolicy cohort was much higher than that in the prepolicy cohort (86.52% vs. 55.67%). An increase in the utilization of bid-winning stents may reduce the cost of PCI for ACS patients and therefore may reduce their financial burden.

This study also revealed that the proportion of ACS patients who had 3 or more coronary stents implanted during their first PCI in the postpolicy cohort was higher than that in the pre-policy cohort (6.28% vs. 4.39%). Although shared decision making between the patient (or his or her legal authorized representative) and the operation physician and the patient's informed consent will be conducted before PCI in Shanghai, the suggestions from the operation physician will have more influence on these processes. The changes in the number of coronary stents implanted in a short period may be due to different reasons, such as the appropriateness of PCI and the weakness of bid-winning stents (such as a greater number of short stents among bid-winning stents) [18, 19]. Considering the greater potential risk of coronary thrombosis and subsequent revascularization in ACS patients with more coronary stents implanted during PCI, the reasons for these changes should be analyzed and addressed in the future.

### No significant adverse effects of policy implementation on the risk of PCI in the short run

The implementation of coronary stent policies in Shanghai may reduce the cost of PCI, but whether this practice will adversely affect the risk of PCI is a major concern [20]. By using Cox

**Table 4. Multivariate Cox models for MACEs and revascularization 1 year after PCI.**

| | | MACEs | | | | | Revascularization | | | | |
|---|---|---|---|---|---|---|---|---|---|---|---|
| | Influencing factors [a] | Estimate | SD | $\chi^2$ | p value | HR[b] | Estimate | SD | $\chi^2$ | p value | HR[b] |
| **Policy implementation** | | -0.2060 | 0.0609 | 11.45 | 0.0007 | 0.814 | -0.1964 | 0.0646 | 9.24 | 0.0024 | 0.822 |
| **Age (years)** | | | | | | | | | | | |
| | 60- | 0.0287 | 0.0739 | 0.15 | 0.6979 | 1.029 | 0.0445 | 0.0777 | 0.33 | 0.5671 | 1.045 |
| | 70- | -0.0982 | 0.0826 | 1.41 | 0.2346 | 0.906 | -0.1434 | 0.0882 | 2.65 | 0.1038 | 0.866 |
| **Male** | | 0.2454 | 0.0760 | 10.43 | 0.0012 | 1.278 | 0.2017 | 0.0803 | 6.31 | 0.0120 | 1.224 |
| **No insurance** | | 0.0440 | 0.0969 | 0.21 | 0.6500 | 1.045 | 0.0595 | 0.1019 | 0.34 | 0.5592 | 1.061 |
| **Previous medical history** | | | | | | | | | | | |
| | Diabetes | 0.2342 | 0.0668 | 12.29 | 0.0005 | 1.264 | 0.2371 | 0.0712 | 11.10 | 0.0009 | 1.268 |
| | Hypertension | 0.1679 | 0.0710 | 5.59 | 0.0180 | 1.183 | 0.1450 | 0.0751 | 3.73 | 0.0535 | 1.156 |
| | Hyperlipidemia | -0.0982 | 0.0725 | 1.83 | 0.1756 | 0.906 | -0.0556 | 0.0770 | 0.52 | 0.4703 | 0.946 |
| | Stroke | -0.0120 | 0.3056 | 0.00 | 0.9686 | 0.988 | -0.0380 | 0.3374 | 0.01 | 0.9105 | 0.963 |
| | Chronic kidney disease | 0.0117 | 0.1034 | 0.01 | 0.9097 | 1.012 | 0.0461 | 0.1101 | 0.18 | 0.6757 | 1.047 |
| | COPD | -0.2480 | 0.1854 | 1.79 | 0.1811 | 0.780 | -0.3875 | 0.2143 | 3.27 | 0.0705 | 0.679 |
| | MI | -0.0464 | 0.1225 | 0.14 | 0.7049 | 0.955 | -0.0940 | 0.1337 | 0.49 | 0.4822 | 0.910 |
| | CABG | -0.6097 | 0.2277 | 7.17 | 0.0074 | 0.544 | -0.8020 | 0.2674 | 9.00 | 0.0027 | 0.448 |
| **Tertiary hospital** | | 0.2170 | 0.0925 | 5.50 | 0.0190 | 1.242 | 0.2854 | 0.1004 | 8.08 | 0.0045 | 1.330 |
| **NYHA or Killip functional classification (reference: I)** | | | | | | | | | | | |
| | IV | 0.2994 | 0.1590 | 3.54 | 0.0598 | 1.349 | 0.0842 | 0.1842 | 0.21 | 0.6476 | 1.088 |
| | II or III | -0.2132 | 0.0660 | 10.42 | 0.0012 | 0.808 | -0.2576 | 0.0706 | 13.31 | 0.0003 | 0.773 |
| **Number of coronary stents implanted (reference: 1)** | | | | | | | | | | | |
| | 2 | 0.0861 | 0.0684 | 1.58 | 0.2081 | 1.090 | 0.1275 | 0.0720 | 3.13 | 0.0768 | 1.136 |
| | 3 | -0.0290 | 0.1448 | 0.04 | 0.8412 | 0.971 | 0.0006 | 0.1526 | 0.00 | 0.9971 | 1.001 |
| | 4 or more | 0.2294 | 0.2923 | 0.62 | 0.4326 | 1.258 | 0.1399 | 0.3200 | 0.19 | 0.6621 | 1.150 |
| **Outpatient medicines used within 1 year after PCI (reference: Other medicine)** | | | | | | | | | | | |
| | Aspirin | -0.1624 | 0.0689 | 5.55 | 0.0184 | 0.850 | -0.1269 | 0.0737 | 2.96 | 0.0853 | 0.881 |
| | Clopidogrel | 0.0088 | 0.0721 | 0.01 | 0.9029 | 1.009 | -0.0018 | 0.0765 | 0.00 | 0.9808 | 0.998 |
| | Metoprolol | -0.1276 | 0.0610 | 4.38 | 0.0364 | 0.880 | -0.0535 | 0.0650 | 0.68 | 0.4100 | 0.948 |
| | Ticagrelor | 0.2719 | 0.0746 | 13.28 | 0.0003 | 1.312 | 0.3033 | 0.0797 | 14.50 | 0.0001 | 1.354 |
| | Atorvastatin | 0.0178 | 0.0753 | 0.06 | 0.8132 | 1.018 | 0.0482 | 0.0804 | 0.36 | 0.5488 | 1.049 |
| | Ezetimibe | 0.0849 | 0.0748 | 1.29 | 0.2565 | 1.089 | 0.1698 | 0.0774 | 4.81 | 0.0283 | 1.185 |
| | Sacubitril/valsartan | 0.3372 | 0.0681 | 24.50 | < .0001 | 1.401 | 0.2170 | 0.0730 | 8.83 | 0.0030 | 1.242 |
| | Trimetazidine dihydrochloride | 0.1763 | 0.1154 | 2.34 | 0.1265 | 1.193 | 0.2757 | 0.1186 | 5.40 | 0.0201 | 1.317 |
| | Pravastatin sodium | -0.0584 | 0.1155 | 0.26 | 0.6130 | 0.943 | 0.0161 | 0.1194 | 0.02 | 0.8926 | 1.016 |
| | Nicorandil | 0.0481 | 0.1219 | 0.16 | 0.6932 | 1.049 | -0.0263 | 0.1339 | 0.04 | 0.8442 | 0.974 |
| | Isosorbide dinitrate | 0.1828 | 0.0793 | 5.32 | 0.0211 | 1.201 | 0.1549 | 0.0844 | 3.37 | 0.0666 | 1.167 |
| | Rosuvastatin | -0.0136 | 0.0819 | 0.03 | 0.8685 | 0.987 | -0.0035 | 0.0867 | 0.00 | 0.9677 | 0.996 |
| **Testing global null hypothesis** | | | | | | | | | | | |
| | Likelihood ratio | | | 163.91 | < .0001 | | | | 158.51 | < .0001 | |
| | $\chi^2$ Wald | | | 162.23 | < .0001 | | | | 151.91 | < .0001 | |

[a] All independent variables in the models were 1–0 variables (1 for "yes", 0 for "no"), n = 6375

[b] HR: hazard ratio.

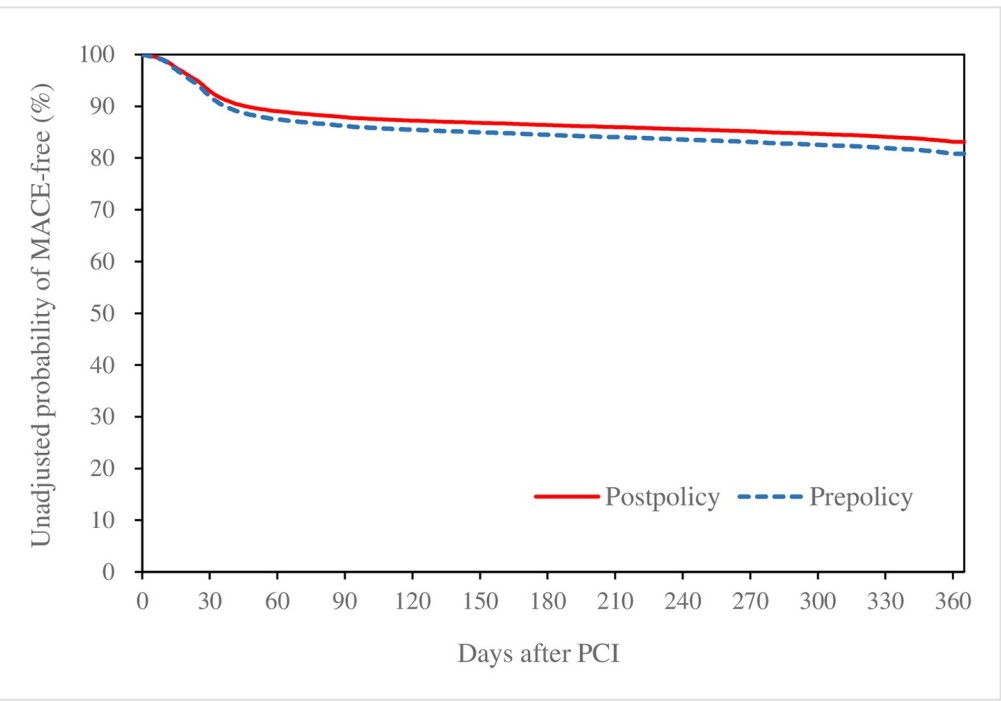

a. Unadjusted

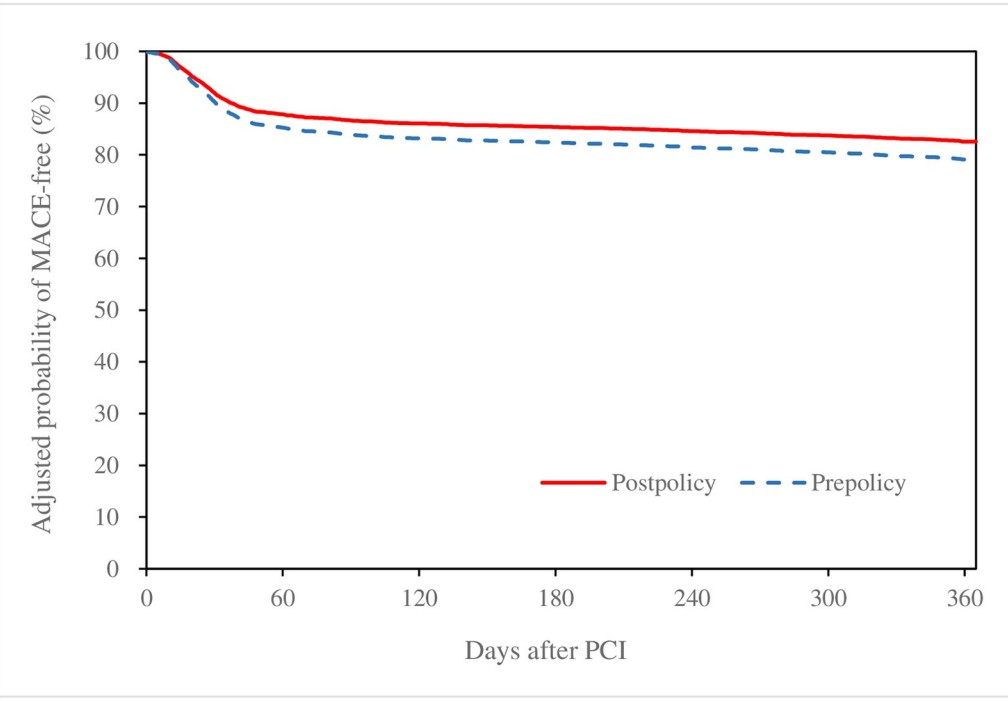

b. Adjusted

**Fig 1. MACE-free survival curves.** a) Unadjusted, b) Adjusted.

models, this study found that the unadjusted and adjusted hazard ratios for MACEs at 1 year after PCI for the postpolicy cohort relative to the prepolicy cohort were 0.869 (p<0.0001) and 0.814 (p = 0.0007), respectively. The implementation of coronary stent policies in Shanghai had no significant adverse effects on the risk of PCI, at least in the short term.

In addition, this study revealed that the reduction in MACEs in the postpolicy cohort compared with the prepolicy cohort was mainly because of the reduction in coronary artery revascularization, with unadjusted and adjusted hazard ratios of 0.873 (p<0.0001) and 0.822 (p = 0.0024), respectively.

Finally, this study did not find that the implementation of coronary stent policies in Shanghai affected the risk of death from any cause, nonfatal MI with no revascularization, or coronary artery restenosis or stent thrombosis at 1 year after PCI.

## Effects of patient characteristics and medications on the risk of PCI

There are many factors that may influence the risk of PCI in ACS patients. Demographic characteristics (such as age and sex) [21], the prevalence of health factors (such as cholesterol, blood pressure, and glucose control) [21, 22], and major medications (such as antiplatelet therapy and statins) [23, 24] are often considered to be influencing factors that may affect the risk of PCI. In this study, we also found that some of the patient demographic and clinical characteristics influenced the risk of PCI. According to the multivariate Cox models, male ACS patients and ACS patients with diabetes or hypertension had a greater risk of MACEs at 1 year after PCI, while ACS patients who had previously undergone CABG had a lower risk of MACEs, and ACS patients with hypertension or severe cardiac impairment (NYHA or Killip functional classification IV) had a greater risk of death at 1 year after PCI.

ACS is in a highly thrombotic state, and the utilization of antiplatelet drugs is vital for preventing thrombotic complications [25]. This study revealed that the use of aspirin after PCI significantly reduced the risk of MACEs and the risk of death at 1 year after PCI after controlling for other factors in multivariate Cox models. In addition, due to the ability of metoprolol to reduce cardiac oxygen consumption and the degree of myocardial ischemia, the use of metoprolol after PCI in ACS patients also reduced the risk of MACEs and the risk of death at 1 year after PCI. However, ticagrelor, sacubitril/valsartan and isosorbide dinitrate are often used in patients who undergo PCI in emergency situations or in patients who have heart failure to reduce the risk of death. The utilization of these medicines by ACS patients after PCI itself indicated a greater probability of experiencing MACEs.

However, it is highly unexpected that the ACS patients who had undergone PCI at tertiary hospitals in Shanghai had a higher risk of MACEs at 1 year after PCI, although cardiac function, number of implanted stents, some patient demographic and clinical characteristics, and medication utilization after PCI were controlled for in the multivariate Cox model. Usually, physicians in tertiary hospitals in Shanghai have greater competence in PCI than do those in secondary hospitals. The reason for the greater risk of MACEs at 1 year after PCI among ACS patients in tertiary hospitals should be explored in the future.

## Limitations

This study was based on real-world data and can reflect the effect of coronary stent policies in Shanghai on the risk of PCI in ACS patients in reality. This study has several limitations that may have led to bias.

First, the risk factors leading to the occurrence of MACEs in ACS patients after PCI were controlled for using multivariate Cox models. However, these factors, such as health behaviors (smoking, physical activity, diet and weight), health factors (cholesterol, blood pressure, and

glucose control), lesion complexity [26, 27], and other unknown confounders, were not fully captured in the models, despite the inclusion of the NYHA or Killip functional classification. Therefore, this study may have had some bias. Second, twelve medicines used in outpatient care were included in the above models, but the dosage and duration of medicine use were not considered in the models. Nonetheless, some meaningful results related to medication use were found from the models. Third, the COVID-19 pandemic obviously affected medical care in Shanghai only in April and May of 2022 when the city was almost shut down (but not including hospitals) and affected only the postpolicy cohort. However, in 1 year after the first PCIs, ACS patients in severe condition (such as coronary artery restenosis, coronary stent thrombosis, MI) generally seek medical care through emergency departments or outpatient departments and inpatient care thereafter, even in April and May of 2022 in Shanghai. Considering that the main reduction in MACEs was coronary artery revascularization between the two cohorts and the number of deaths included those who died at hospitals and at homes, the COVID-19 pandemic might have limited the ability of the study to confirm these findings. Fourth, this study involved ACS patients who had undergone PCI in Shanghai, and the implementation of coronary stent policies involved Shanghai's local supportive measures. Generalizing our results to other regions of China or other countries should be performed with caution. Fifth, this study assessed the effect of these policies on the risk of PCI in ACS patients at 1 year; therefore, the long-term effect of these policies on the risk of PCI should be considered in the next few years.

## Conclusions

The implementation of coronary stent policies changed coronary stent utilization but had no significant adverse effect on the risk of PCI in ACS patients in the short run. The hazard ratio for MACEs at 1 year after PCI in ACS patients in the postpolicy cohort relative to the prepolicy cohort was 0.869. The study findings provide important information for policy-makers as well as clinicians regarding the effect of policy implementation on the safety of PCI and may serve as a response to public concerns to some extent. However, the reasons for changes in the number of coronary stents implanted during PCI should be analyzed and addressed in the future.

## Supporting information

**S1 Checklist. Human participants research checklist.**
(DOCX)

**S1 Table. Cox model for restenosis or stent thrombosis 1 year after PCI.**
(DOCX)

**S2 Table. Cox model for MI 1 year after PCI.**
(DOCX)

**S3 Table. Cox model for death 1 year after PCI.**
(DOCX)

## Acknowledgments

We gratefully acknowledge the significant contributions of Wenqi Tian and Jiazhen Liu from the Shanghai Statistics Center for Health in helping us with the data extraction in this study.

## Author Contributions

**Conceptualization:** Zhenyi Shao, Dongzhe Lu, Di Xue.

**Data curation:** Zhenyi Shao, Yue Wang.

**Formal analysis:** Dongzhe Lu, Di Xue.

**Project administration:** Di Xue.

**Writing – original draft:** Zhenyi Shao, Dongzhe Lu.

**Writing – review & editing:** Yue Wang, Di Xue.

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
