## [Decision Letter · Decision Letter 0]

12 Dec 2023

PONE-D-23-30421The Effect of Coronary Stent Policies on the Risks of PCI in ACS Patients in Shanghai: Real-world EvidencePLOS ONE

Dear Dr. Xue,

Thank you for submitting your manuscript to PLOS ONE. After careful consideration, we feel that it has merit but does not fully meet PLOS ONE’s publication criteria as it currently stands. Therefore, we invite you to submit a revised version of the manuscript that addresses the points raised during the review process.

Please address the points raised by the reviewers below;

We look forward to receiving your revised manuscript.

Kind regards,

Shukri AlSaif

Academic Editor

PLOS ONE

Journal Requirements:

2. Please include a separate caption for each figure in your manuscript.

Reviewers' comments:

Reviewer's Responses to Questions

**Comments to the Author**

1. Is the manuscript technically sound, and do the data support the conclusions?

Reviewer #1: Yes

Reviewer #2: No

2. Has the statistical analysis been performed appropriately and rigorously? 

Reviewer #1: Yes

Reviewer #2: No

3. Have the authors made all data underlying the findings in their manuscript fully available?

Reviewer #1: Yes

Reviewer #2: Yes

4. Is the manuscript presented in an intelligible fashion and written in standard English?

Reviewer #1: No

Reviewer #2: Yes

5. Review Comments to the Author

Reviewer #1: This is an interesting manuscript whose aim is to provide an analysis of the effects of coronary stent policies on the risks of percutaneous coronary intervention (PCI) in acute coronary syndrome (ACS) inpatients based on real-world data in China. The manuscript is quite well written, despite sometimes it is not as fluent as expected. Notwithstanding, there are some issues that I suggest to review:

- the title is not that clear, so please refine it;

- the stent differences between pre- and postpolicy should be explained in more depth;

- details about the responsibile for stent selection (single operator, chief of the cathlab, chief of the whole cardiology division, etc...) as well as the data collection responsible should be provided too;

- the reason why prasugrel has not been mentioned in the methods should be stated;

- data about cholesterol target achievement after the first ACS event should be included, if possible;

- stent thrombosis underlying causes should be mentioned;

- a deeper speculation about all the potential factors responsible for MACEs 1 year after PCI could make the discussion more interesting;

- in the end, what about the choice to implant more than three stents in the postpolicy era? Could the authors speculate about both the thrombotic risk of this choice and the subsequent increase in re-hospitalization rate for further revascularization?

Reviewer #2: The authors concluded that the implementation of coronary stent policies did not adversely affect the risks of PCI in ACS patients. Unfortunately, many important data were not evaluated in the analysis. The results obtained in the study were not supported if there were no such variables.

Specific comments:

1. In the study, ACS included UAP, STEMI and NSTEMI. Each number of patients should be added. Clinical outcomes might significantly differ among etiologies.

2. Lesional characteristics and complications during PCI as well as both D2B time and myocardial damages in MI patients significant affect mortality at 1-year. However, there are no data on such variables. They are much more important than types of stents.

3. During the study, COVID-19 pandemic affected the medical situations. It does not be evaluated.

4. Tables 1 and 2：Does MI mean past history before index PCI? Please clarify.

5. Definitions of risk factors should be added. Proportions of some co-morbidities such as CKD were surprisingly low.

6. PLOS authors have the option to publish the peer review history of their article (what does this mean?). If published, this will include your full peer review and any attached files.

Reviewer #1: **Yes: **Iacovelli Fortunato

Reviewer #2: No

---

## [Author Response · Author response to Decision Letter 0]

2 Feb 2024

Review Comments to the Author

Reviewer #1: This is an interesting manuscript whose aim is to provide an analysis of the effects of coronary stent policies on the risks of percutaneous coronary intervention (PCI) in acute coronary syndrome (ACS) inpatients based on real-world data in China. The manuscript is quite well written, despite sometimes it is not as fluent as expected. Notwithstanding, there are some issues that I suggest to review:

Response: Thank you for the comments. We have revised the manuscript and hired a company (American Journal of Experts, AJE) to help us check the English writing again.

- the title is not that clear, so please refine it;

Response: We revised the title to “ The Effect of Coronary Stent Policies on the Risk of Percutaneous Coronary Intervention among Acute Coronary Syndrome Patients in Shanghai: Real-world Evidence” to enhance the clarity.

- the stent differences between pre- and postpolicy should be explained in more depth;

Response: In the ACS patients who underwent PCI for the first time, all the bid-winning stents implanted were drug-eluting or drug-coating stents, while the bid-nonwinning stents included drug-eluting, drug-coating, and biodegradable stents. Bare metal stents were not used in any of the patients. The proportions of drug-eluting stents, drug-coated stents, biodegradable stents and unknown types of stents implanted were 78.90%, 19.75%, 0.29% and 1.06%, respectively, in the postpolicy cohort, while these proportions were 71.07%, 21.90%, 4.51% and 2.52%, respectively, in the prepolicy cohort. More drug-coated stents and biodegradable stents were used in the prepolicy cohort (χ2= 600.69, p<0.0001). 

We have added the above information to the manuscript. (See lines 163-170 on pages 11.)

- details about the responsibile for stent selection (single operator, chief of the cathlab, chief of the whole cardiology division, etc...) as well as the data collection responsible should be provided too;

Response: We have revised the corresponding description to clarify the relevant statements. (see lines 231-234 on page 18.)

- the reason why prasugrel has not been mentioned in the methods should be stated;

Response: Because vascular endothelial injury during PCI leads to stent thrombosis, dual antiplatelet therapy (DAPT) with aspirin plus a P2Y12 inhibitor is the standard-of-care treatment for patients undergoing PCI [16]. According to the Chinese guidelines for cardiovascular disease prevention (2017), aspirin plus clopidogrel were recommended for use for at least 12 months after PCI; for patients who cannot tolerate clopidogrel or have shown clear evidence of clopidogrel resistance, ticagrelor or prasugrel were recommended as alternative medicines [17]. However, there were no ACS patients who used prasugrel after PCI in our study; therefore, it was not included in the analyses.

We have added the above information to the Methods section. (see lines 107-114 on page 6.)

- data about cholesterol target achievement after the first ACS event should be included, if possible;

Response: Because the data we used for the analyses were from real-world data and because the cholesterol test results could not be extracted from the databases or did not exist in some cases, we have no data about cholesterol target achievement after the first ACS event.

We have added information about various health factors (e.g., cholesterol, blood pressure, and glucose control) as an example of risk factors in the limitations section. (see line 288 on page 21.)

- stent thrombosis underlying causes should be mentioned;

Response: We have added information to the methods section about how vascular endothelial injury during PCI that can lead to stent thrombosis. We also added information about the use of dual antiplatelet therapy (DAPT) for prevention. Furthermore, we described physical activity as examples of risk factors in the limitations section. (see lines 107-113 on page 6 and 288 on page 21.)

- a deeper speculation about all the potential factors responsible for MACEs 1 year after PCI could make the discussion more interesting;

Response: Thank you for your suggestions. We have also added health behaviors (smoking, physical activity, diet and weight) as examples of risk factors in the limitations section. (see lines 287-288 on page 21.)

- in the end, what about the choice to implant more than three stents in the postpolicy era? Could the authors speculate about both the thrombotic risk of this choice and the subsequent increase in re-hospitalization rate for further revascularization?

Response: Thank you for your suggestions. We have added information about these issues to the discussion section.

“Considering the greater potential risk of coronary thrombosis and subsequent revascularization in ACS patients with more coronary stents implanted during PCI, the reasons for these changes should be analyzed and addressed in the future.” (see lines 237-239 on page 18.)

Reviewer #2: The authors concluded that the implementation of coronary stent policies did not adversely affect the risks of PCI in ACS patients. Unfortunately, many important data were not evaluated in the analysis. The results obtained in the study were not supported if there were no such variables.

Response: Thank you for the comments. Because the study was based on real-world data, some data related to risk factors could not be extracted from the database. According to the comments and suggestions, we have added more description to the limitations section and revised some of the descriptions to clarify the conclusion.

1) Changed subtitle: No significant adverse effects of policy implementation on the risk of PCI in the short run. (see lines 240-241 on page 18.)

2) Changed description: The implementation of coronary stent policies in Shanghai had no significant adverse effect on the risk of PCI, at least in the short term. (see lines 246-247 on page 19.)

3) Changed conclusion: The implementation of coronary stent policies changed coronary stent utilization but had no significant adverse effect on the risk of PCI in ACS patients in the short run. (see lines 309-310 on page 22.)

Specific comments:

1. In the study, ACS included UAP, STEMI and NSTEMI. Each number of patients should be added. Clinical outcomes might significantly differ among etiologies.

Response: According to ICD-10 codes, we added the comparison of the diagnostic classification between the postpolicy and prepolicy cohorts. The result showed that there was no significant difference in diagnostic classification of ACS (unstable angina pectoris, ST-elevation MI, non ST-elevation MI or undetermined) between the two cohorts. 

We have revised the description about diagnostic classification of ACS in the Methods section and added corresponding information in the Results section and Table 1. (see lines 86-87 on page 5 and lines 152-156 on pages 8-9.)

2. Lesional characteristics and complications during PCI as well as both D2B time and myocardial damages in MI patients significant affect mortality at 1-year. However, there are no data on such variables. They are much more important than types of stents.

Response: Because the study was based on real-world data, some data related to risk factors, such as lesion characteristics, D2B duration and test results for myocardial damage, could not be extracted from the database; these factors could not be controlled for in the study.

Considering that lesion complexity was not controlled for in the present study, this was mentioned as a limitation. In addition, we included the NYHA or Killip functional classification in the present study, which could reflect myocardial damage to some extent.

There are several explanations for D2B duration, and additional studies are needed to determine the causal relationship between reduced D2B duration and improved patient prognosis. In China, 90 minutes of D2B therapy were required for thrombolysis and/or PCI in the emergency care of ACS patients by the central and local governments long before 2019.

PCI is a mature procedure, but bleeding at the puncture site, vascular injury, allergy to contrast media and stent thrombosis after the procedure are the main complications, and coronary artery dissection and perforation rarely occur during PCI. In this study, we focused on MACEs caused by PCI at 1 year, and MACEs themselves can reflect vascular injuries during and after PCI. Therefore, we did not include the issue of complications during PCI in this study, as did many other similar studies.

3. During the study, COVID-19 pandemic affected the medical situations. It does not be evaluated.

Response: In the present study, inpatients with a first diagnosis of ACS who had undergone first-time PCI were included; these patients were Shanghai residents discharged between March 1, 2019, and April 30, 2022, and by Aug 13, 2022, 1 year, at least, had passed after their first-time PCI. The COVID-19 pandemic obviously affected medical care in Shanghai only in April and May of 2022 when the city was almost shut down (but not including hospitals) and affected only the postpolicy cohort. However, in 1 year after the first PCIs, ACS patients in severe condition (such as coronary artery restenosis, coronary stent thrombosis, MI) generally seek medical care through emergency departments or outpatient departments and inpatient care thereafter, even in April and May of 2022 in Shanghai. Considering that the main reduction in MACEs was coronary artery revascularization between the two cohorts and the number of deaths included those who died at hospitals and at homes, the COVID-19 pandemic might have limited the ability of the study to confirm these findings.

Thank you for the suggestions. We have added above information (“by Aug 13, 2022, 1 year, at least,”) to the inclusion criteria in the Methods section for more clarity and added the issue of COVID-19 pandemic to the limitations section. (see line 77 on page 4 and lines 293-302 on pages 21.)

4. Tables 1 and 2：Does MI mean past history before index PCI? Please clarify.

Response: For more clarity, the indicators of patient characteristics and medications have been bolded, and “Medical history” has been revised to “Previous medical history” in Table 1. We have also revised the format of Table 4, and “Medical history” has been revised to “Previous medical history” in Table 4. (See page 9 and page 14.)

5. Definitions of risk factors should be added. Proportions of some co-morbidities such as CKD were surprisingly low.

Response: The data related to comorbidities in the study were extracted from the database of patient medical records. The exact reasons for the low or high incidence of comorbidities in ACS patients in Shanghai are unknown. Compared to the study of FAVOR III conducted among patients undergoing PCI who had at least one lesion with a diameter stenosis of 50–90% in a coronary artery in China, the comorbidities and previous medical history of diabetes, hypertension, hyperlipidemia and previous MI were similar or slightly lower in our study, while previous stroke was lower and previous CABG was greater in our study. Shanghai has higher quality health care in China, which may lead to differences in patient general health status.

We have added definitions of the risk factors to the limitations section. (see line 286 on page 21.)

 6. PLOS authors have the option to publish the peer review history of their article (what does this mean?). If published, this will include your full peer review and any attached files.

Do you want your identity to be public for this peer review? For information about this choice, including consent withdrawal, please see our Privacy Policy.

Reviewer #1: Yes: Iacovelli Fortunato

Reviewer #2: No

Journal Requirements:

Response: We have checked and revised the manuscript to meet the style requirements of PLOS ONE, including those for file naming.

2. Please include a separate caption for each figure in your manuscript.

Response: We have added a separate caption for the figure in our manuscript.

Response: We have added captions for our Supporting Information files at the end of our manuscript.

---

## [Decision Letter · Decision Letter 1]

17 Mar 2024

The Effect of Coronary Stent Policies on the Risk of Percutaneous Coronary Intervention among Acute Coronary Syndrome Patients in Shanghai: Real-world Evidence

PONE-D-23-30421R1

Dear Dr.Xue,

We are  pleased to inform you that your manuscript has been judged scientifically suitable for publication and will be formally accepted for publication once it meets all outstanding technical requirements.

Kind regards,

Shukri AlSaif

Academic Editor

PLOS ONE

Additional Editor Comments (optional):

I have evaluated the revised manuscript and consider the revisions the authors made sufficient for acceptance even though one of the reviewers suggested rejection.

Reviewers' comments:

**Comments to the Author**

1. If the authors have adequately addressed your comments raised in a previous round of review and you feel that this manuscript is now acceptable for publication, you may indicate that here to bypass the “Comments to the Author” section, enter your conflict of interest statement in the “Confidential to Editor” section, and submit your "Accept" recommendation.

Reviewer #1: All comments have been addressed

Reviewer #2: (No Response)

2. Is the manuscript technically sound, and do the data support the conclusions?

Reviewer #1: Yes

Reviewer #2: No

3. Has the statistical analysis been performed appropriately and rigorously? 

Reviewer #1: Yes

Reviewer #2: I Don't Know

4. Have the authors made all data underlying the findings in their manuscript fully available?

Reviewer #1: Yes

Reviewer #2: Yes

5. Is the manuscript presented in an intelligible fashion and written in standard English?

Reviewer #1: Yes

Reviewer #2: Yes

6. Review Comments to the Author

Please use the space provided to explain your answers to the questions above. 

Reviewer #1: (No Response)

Reviewer #2: Some comments raised by the reviewer have been resolved. However, the reviewer ia not satisfied with important issues

Once again, many important data were not evaluated in the analysis.

The authors agree that the study was based on real-world data, some data related to risk factors could not be extracted from the database. If so, the results cannot support the conclusion.

Many previous reports suggest that clinical prognosis and outcomes significantly differ among patients with STEMI, NSTEMI, and UAP. Athough the authors added data on the number of each category, there was no data on evaluation of clinical outcomes among them.

Complications during PCI significantly affect larger infarction size, resulting in worse clinical outcomes.

7. PLOS authors have the option to publish the peer review history of their article (what does this mean?). If published, this will include your full peer review and any attached files.

Reviewer #1: **Yes: **Iacovelli Fortunato

Reviewer #2: **Yes: **Hideki ISHII

---

## [Editor Report · Acceptance letter]

22 Mar 2024

PONE-D-23-30421R1 

PLOS ONE

Dear Dr. Xue, 

I'm pleased to inform you that your manuscript has been deemed suitable for publication in PLOS ONE. Congratulations! Your manuscript is now being handed over to our production team.

Kind regards, 

on behalf of

Dr. Shukri AlSaif 

Academic Editor

PLOS ONE